# Towards visible soliton microcomb generation

Seung Hoon Lee[1], Dong Yoon Oh [1], Qi-Fan Yang[1], Boqiang Shen[1], Heming Wang[1], Ki Youl Yang [1], Yu-Hung Lai[1], Xu Yi[1], Xinbai Li[1,2] & Kerry Vahala[1]

Frequency combs have applications that extend from the ultra-violet into the mid-infrared bands. Microcombs, a miniature and often semiconductor-chip-based device, can potentially access most of these applications, but are currently more limited in spectral reach. Here, we demonstrate mode-locked silica microcombs with emission near the edge of the visible spectrum. By using both geometrical and mode-hybridization dispersion control, devices are engineered for soliton generation while also maintaining optical $Q$ factors as high as 80 million. Electronics-bandwidth-compatible (20 GHz) soliton mode locking is achieved with low pumping powers (parametric oscillation threshold powers as low as 5.4 mW). These are the shortest wavelength soliton microcombs demonstrated to date and could be used in miniature optical clocks. The results should also extend to visible and potentially ultra-violet bands.

[1] T. J. Watson Laboratory of Applied Physics, California Institute of Technology, Pasadena, CA 91125, USA. [2] State Key Laboratory of Advanced Optical Communication Systems and Networks, School of Electronics Engineering and Computer Science, Peking University, Beijing 100871, China. Seung Hoon Lee, Dong Yoon Oh, Qi-Fan Yang, Boqiang Shen, and Heming Wang contributed equally to this work. Correspondence and requests for materials should be addressed to K.V. (email: vahala@caltech.edu)

Soliton mode locking[1–5] in frequency microcombs[6] provides a pathway to miniaturize many conventional comb applications. It has also opened investigations into new nonlinear physics associated with dissipative Kerr solitons[1] and Stokes solitons[7]. In contrast to early microcombs[6], soliton microcombs eliminate instabilities, provide stable (low-phase-noise) mode locking, and feature a highly reproducible spectral envelope. Many applications of these devices are being studied, including chip-based optical frequency synthesis[8], secondary time standards[9], and dual-comb spectroscopy[10–12]. Also, a range of operating wavelengths is opening up by use of several low-optical-loss dielectric materials for resonator fabrication. In the near-infrared (IR), microcombs based on magnesium fluoride[1], silica[2,13], and silicon nitride[3–5,14,15] are being studied for frequency metrology and frequency synthesis. In the mid-IR spectral region silicon nitride[16], crystalline[17], and silicon-based[18] Kerr microcombs, as well as quantum-cascade microcombs[19] are being studied for application to molecular fingerprinting.

At shorter wavelengths below 1 μm microcomb technology would benefit optical atomic clock technology[20], particularly efforts to miniaturize these clocks. For example, microcomb optical clocks based on the D1 transition (795 nm) and the two-photon clock transition[21] (798 nm) in rubidium have been proposed[9,22]. Also, a microcomb clock using two-point locking to rubidium D1 and D2 lines has been demonstrated[23] by frequency doubling from the near-IR. More generally, microcomb sources in the visible and ultra-violet bands could provide a miniature alternative to larger mode-locked systems such as titanium sapphire lasers in cases where high power is not required. It is also possible that these shorter wavelength systems could be applied in optical coherence tomography systems[24–26]. Efforts directed toward short wavelength microcomb operation include 1 μm microcombs in silicon nitride microresonators[27] as well as harmonically generated combs. The latter have successfully converted near-IR comb light to shorter wavelength bands[28] and even into the visible band[29,30] within the same resonator used to create the initial comb of near-IR frequencies. Also, crystalline

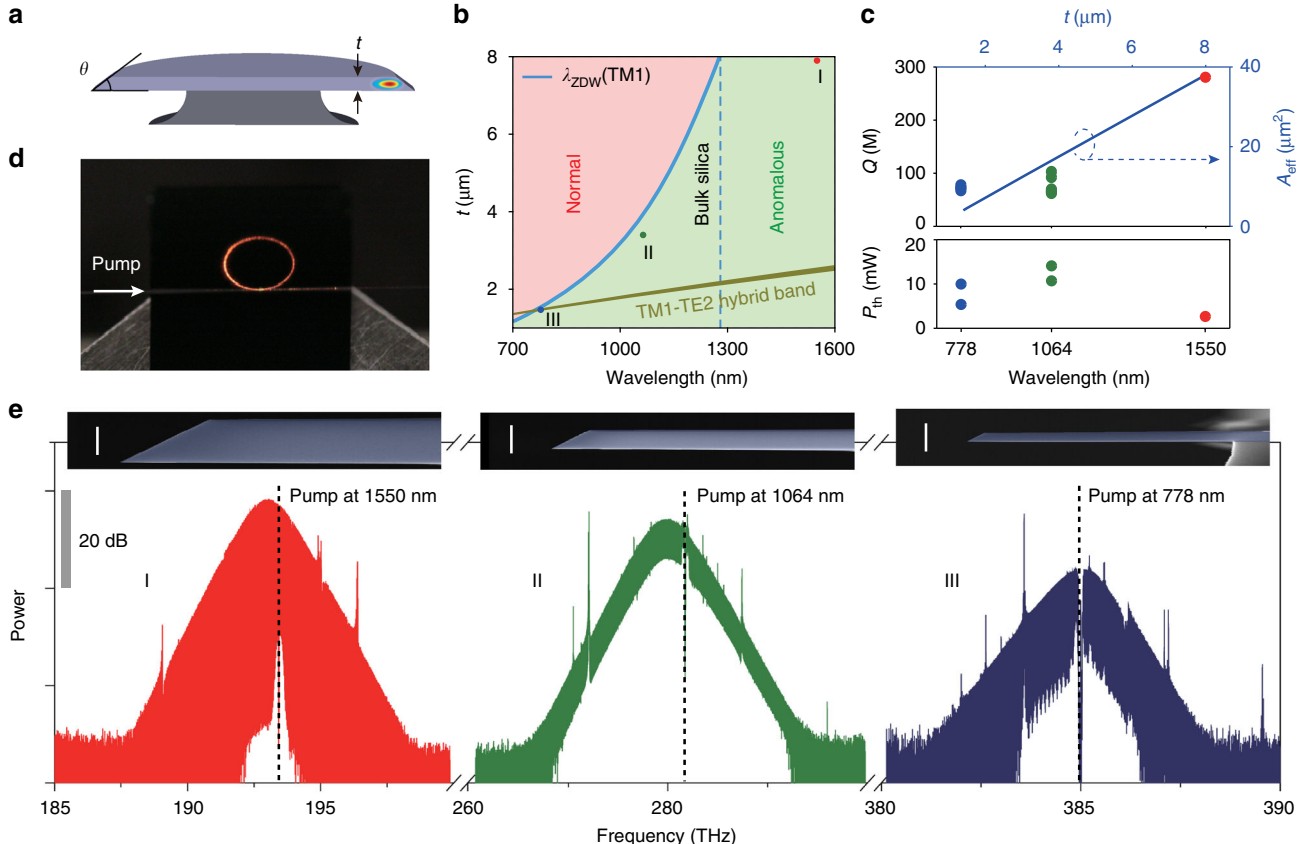

**Fig. 1** Soliton frequency comb generation in dispersion-engineered silica resonators. **a** A rendering of a silica resonator with the calculated TM1 mode profile superimposed. **b** Regions of normal and anomalous dispersion are shown vs. silica resonator thickness ($t$) and pump wavelength. The zero dispersion wavelength ($\lambda_{ZDW}$) for the TM1 mode appears as a blue curve. The dark green band shows the 10-dB bandwidth of anomalous dispersion created by TM1-TE2 mode hybridization. The plot is made for a 3.2-mm diameter silica resonator with a 40° wedge angle. Three different device types I, II, and III (corresponding to $t = 7.9$, 3.4, and 1.5 μm) are indicated for soliton generation at 1550, 1064, and 778 nm. **c** Measured $Q$ factors and parametric oscillation threshold powers vs. thickness and pump wavelength for the three device types. Powers are measured in the tapered fiber coupler under critical coupling. Effective mode area ($A_{eff}$) of the TM1 mode family is also plotted as a function of wavelength and thickness. **d** A photograph of a silica resonator (Type III device pumped at 778 nm) while generating a soliton stream. The pump light is coupled via a tapered fiber from the left side of the resonator. The red light along the circumference of the resonator and at the right side of the taper is believed to result from short wavelength components of the soliton comb. **e** Soliton frequency comb spectra measured from the devices. The red, green, and blue soliton spectra correspond to device types I, II, and III designed for pump wavelengths 1550, 1064, and 778 nm, respectively. Pump frequency location is indicated by a dashed vertical line. The soliton pulse repetition rate of all devices is about 20 GHz. Differences in SNR of the spectra originate from the resolution of the optical spectrum analyzer (OSA). In particular, the 778 nm comb spectrum was measured using the second-order diffracted spectrum of the OSA, while other comb spectra were measured as first-order diffracted spectra. Insets: cross-sectional SEM images of the fabricated resonators. White scale bar is 5 μm

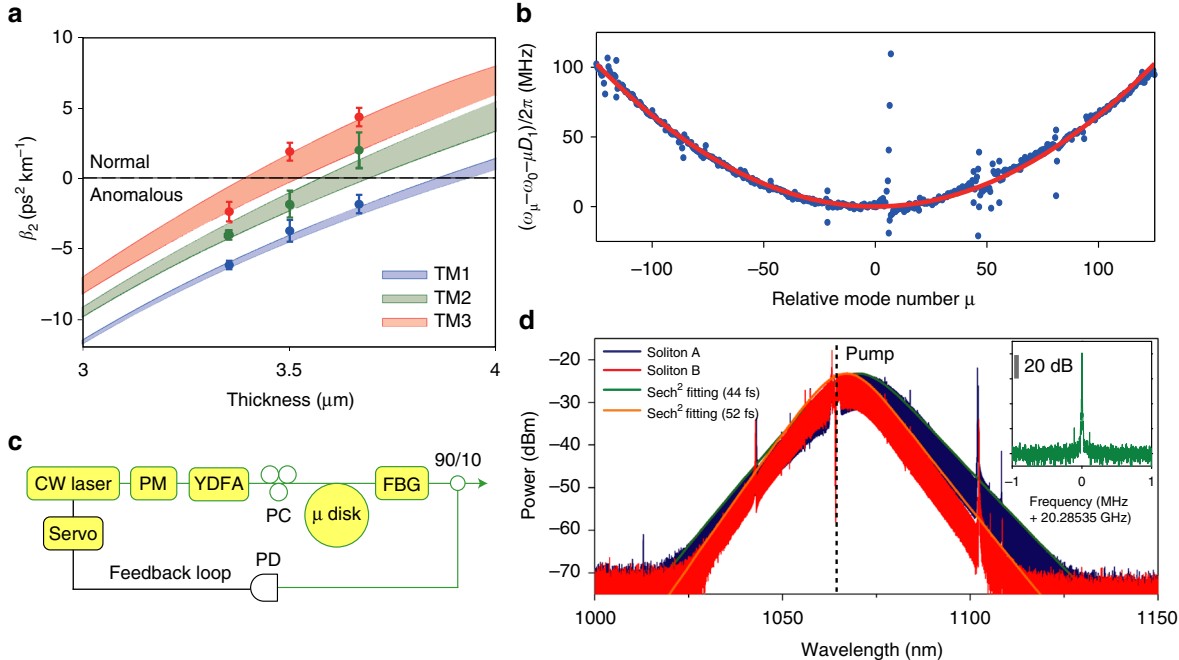

**Fig. 2** Microresonator dispersion engineering and soliton generation at 1064 nm. **a** Simulated GVD of TM mode families vs. resonator thickness. The angle of the wedge ranges from 30° to 40° in the colored regions. Measured data points are indicated and agree well with the simulation. The error bars depict standard deviations obtained from measurement of eight samples having the same thickness. **b** Measured relative mode frequencies (blue points) plotted vs. relative mode number of a soliton-forming TM1 mode family in a 3.4 μm thick resonator. The red curve is a parabolic fit yielding $D_2/2\pi = 3.3$ kHz. **c** Experimental setup for soliton generation. A CW fiber laser is modulated by an electro-optic PM before coupling to a YDFA. The pump light is then coupled to the resonator using a tapered fiber. Part of the comb power is used to servo-control the pump laser frequency. **d** Optical spectra of solitons at 1064 nm generated from the mode family shown in **b**. The two soliton spectra correspond to different power levels with the blue spectrum being a higher power and wider bandwidth soliton. The dashed vertical line shows the location of the pump frequency. The solid curves are $\text{sech}^2$ fittings. Inset: typical detected electrical beatnote showing soliton repetition rate. The weak sidebands are induced by the feedback loop used to stabilize the soliton. The resolution bandwidth is 1 kHz. FBG fiber Bragg grating, PD photodetector, PC polarization controller

resonators[31] and silica microbubble resonators[32] have been dispersion-engineered for comb generation in the 700 nm band. Finally, diamond-based microcombs afford the possibility of broad wavelength coverage[33]. However, none of the short wavelength microcomb systems have so far been able to generate stable mode-locked microcombs as required in all comb applications.

A key impediment to mode-locked microcomb operation at short wavelengths is material dispersion associated with the various dielectric materials used for microresonator fabrication. At shorter wavelengths, these materials feature large normal dispersion that dramatically increases into the visible and ultraviolet bands. While dark soliton pulses can be generated in a regime of normal dispersion[34], bright solitons require anomalous dispersion. Dispersion engineering by proper design of the resonator geometry[22,31,32,35–41] offers a possible way to offset the normal dispersion. Typically, by compressing the waveguide dimension of a resonator, geometrical dispersion will ultimately compensate a large normal material dispersion component to produce overall anomalous dispersion. For example, in silica, strong confinement in bubble resonators[32] and straight waveguides[42] has been used to push the anomalous dispersion transition wavelength from the near-IR into the visible band. Phase matching to ultra-violet dispersive waves has also been demonstrated using this technique[42]. However, to compensate the rising material dispersion this compression must increase as the operational wavelength is decreased, and as a side effect, highly confined waveguides tend to suffer increased optical losses. This happens because mode overlap with the dielectric waveguide interface is greater with reduced waveguide cross-section. Consequently, the residual fabrication-induced roughness of that interface degrades the resonator Q factor and increases pumping power (e.g., comb threshold power varies inverse quadratically with Q factor[43]).

Minimizing material dispersion provides one way to ease the impact of these constraints. In this sense, silica offers an excellent material for short wavelength operation, because it has the lowest dispersion among all on-chip integrable materials. For example, at 778 nm, silica has a group velocity dispersion (GVD) equal to 38 ps² km⁻¹, which is over five times smaller than the GVD of silicon nitride at this wavelength (>200 ps² km⁻¹)[44]. Other integrable materials that are also transparent in the visible, such as diamond[33] and aluminum nitride[45], have dispersion that is similar to or higher than silicon nitride. Silica also features a spectrally broad low-optical-loss window so that optical Q factors can be high at short wavelengths. Here, we demonstrate soliton microcombs with pump wavelengths of 1064 and 778 nm. These are the shortest soliton microcomb wavelengths demonstrated to date. By engineering geometrical dispersion and by employing mode hybridization, a net anomalous dispersion is achieved at these wavelengths while also maintaining high optical Q factors (80 million at 778 nm, 90 million at 1064 nm). The devices have large (millimeter-scale) diameters and produce single soliton pulse streams at rates that are both detectable and processable by low-cost electronic circuits. Besides illustrating the flexibility of silica for soliton microcomb generation across a range of short wavelengths, these results are relevant to potential secondary time standards based on transitions in rubidium[9,22,23]. Using dispersive-wave engineering in silica it might also be possible to extend the emission of these combs into the ultra-violet as recently demonstrated in compact silica waveguides[42].

## Results

**Silica resonator design**. The silica resonator used in this work is shown schematically in Fig. 1a. A fundamental mode profile is overlaid onto the cross-sectional rendering. The resonator design is a variation on the wedge resonator[46], and its geometry can be fully characterized by its resonator diameter, silica thickness ($t$), and wedge angle ($\theta$) (see Fig. 1a). The diameter of all-resonators in this work (and the assumed diameter in all simulations) is 3.2 mm, which corresponds to a free spectral range (FSR) of approximately 20 GHz, and the resonator thickness is controlled to obtain net anomalous dispersion at the design wavelengths, as described in detail below. Further details on fabrication are given elsewhere[46]. As an aside, we note that a waveguide-integrated version of this design is also possible[47]. Adaptation of that device using the methods described here would enable full integration with other photonic elements on the silicon chip.

Figure 1b illustrates how the geometrical dispersion induced by varying resonator thickness $t$ offsets the material dispersion. Regions of anomalous and normal dispersion are shown for the TM1 mode family of a resonator having a wedge angle of 40°. The plots show that thinner resonators enable shorter wavelength solitons. Accordingly, three device types (I, II, and III shown as the colored dots in Fig. 1b) are selected for soliton frequency comb operation at three different pump wavelengths. At a pump wavelength of 1550 nm, the anomalous dispersion window is wide because bulk silica possesses anomalous dispersion at wavelengths above 1270 nm. For this type I device, a 7.9-μm thickness was used. Devices of type II and III have thicknesses near 3.4 and 1.5 μm for operation with pump wavelengths of 1064 and 778 nm, respectively. Beyond geometrical control of dispersion, the type III design also uses mode hybridization to substantially boost the anomalous dispersion. This hybridization occurs within a relatively narrow wavelength band which tunes with $t$ (darker green region in Fig. 1b) and is discussed in detail below. Measured $Q$ factors for the three device types are plotted in the upper panel of Fig. 1c. Maximum $Q$ factors at thicknesses which also produce anomalous dispersion were: 280 million (Type I, 1550 nm), 90 million (Type II, 1064 nm), and 80 million (Type III, 778 nm).

Using these three designs, soliton frequency combs were successfully generated with low threshold pump power. Shown in Fig. 1d is a photograph of a type III device under conditions where it is generating solitons. Figure 1e shows optical spectra of the soliton microcombs generated for each device type. A slight Raman-induced soliton self-frequency-shift is observable in the type I and type II devices[2,48–50]. The pulse width of the type III device is longer and has a relatively smaller Raman shift, which is consistent with theory[50]. The presence of a dispersive wave in this spectrum also somewhat offsets the smaller Raman shift[3]. Scanning electron microscope (SEM) images appear as insets in Fig. 1e and provide cross-sectional views of the three device types. It is worthwhile to note that microcomb threshold power, expressed as $P_{th} \sim A_{eff}/\lambda_P Q^2$ ($\lambda_P$ is pump wavelength and $A_{eff}$ is effective mode area) remains within a close range of powers for all devices (lower panel of Fig. 1c). This can be understood to result from a partial compensation of reduced $Q$ factor in the shorter wavelength devices by reduced optical mode area (see plot in Fig. 1c). For example, from 1550 to 778 nm the mode area is reduced by roughly a factor of 9 and this helps to offset a three times decrease in $Q$ factor. The resulting $P_{th}$ increase (5.4 mW at 778 nm vs. approximately 2.5 mW at 1550 nm) is therefore caused primarily by the decrease in pump wavelength $\lambda_P$. In the following sections additional details on the device design, dispersion, and experimental techniques used to generate these solitons are presented.

**Soliton generation at 1064 nm**. Dispersion simulations for TM modes near 1064 nm are presented in Fig. 2a and show that TM modes with anomalous dispersion occur in silica resonators having oxide thicknesses less than 3.7 μm. Aside from the thickness control, a secondary method to manipulate dispersion is by changing the wedge angle (see Fig. 2a). Both thickness and wedge angle are well controlled in the fabrication process[41]. Precise thickness control is possible because this layer is formed through calibrated oxidation of the silicon wafer. Wedge angles between 30° and 40° were chosen in order to maximize the $Q$ factors[46]. The resonator dispersion is characterized by measuring mode frequencies using a scanning external-cavity diode laser (ECDL) whose frequency is calibrated using a Mach–Zehnder interferometer. As described elsewhere[1,2] the mode frequencies, $\omega_\mu$, are Taylor expanded as $\omega_\mu = \omega_0 + \mu D_1 + \mu^2 D_2/2 + \mu^3 D_3/6$, where $\omega_0$ denotes the pumped mode frequency, $D_1/2\pi$ is the FSR, and $D_2$ is proportional to the GVD, $\beta_2$ ($D_2 = -cD_1^2\beta_2/n_0$, where $c$ and $n_0$ are the speed of light and material refractive index). $D_3$ is a third-order expansion term that is sometimes necessary to adequately fit the spectra (see discussion of 778 nm soliton below). The measured frequency spectrum of the TM1 mode family in a 3.4 μm thick resonator is plotted in Fig. 2b. The plot gives the frequency as relative frequency (i.e., $\omega_\mu - \omega_0 - \mu D_1$) to make clear the second-order dispersion contribution. The frequencies are measured using a radio-frequency calibrated Mach–Zehnder interferometer having a FSR of approximately 40 MHz. Also shown is a fitted parabola (red curve) revealing $D_2/2\pi = 3.3$ kHz (positive parabolic curvature indicates anomalous dispersion). Some avoided mode crossings are observed in the spectrum. The dispersion measured in resonators of different thicknesses, marked as solid dots in Fig. 2a, is in good agreement with numerical simulations.

The experimental setup for generation of 1064 nm pumped solitons is shown in Fig. 2c. The microresonator is pumped by a continuous wave (CW) laser amplified by a ytterbium-doped fiber amplifier (YDFA). The pump light and comb power are coupled to and from the resonator by a tapered fiber[51,52]. Typical pumping power is around 100 mW. Solitons are generated while scanning the laser from higher frequencies to lower frequencies across the pump mode[1–3]. The pump light is modulated by an electro-optic phase modulator (PM) to overcome the thermal transient during soliton generation[2,3,53]. A servo control referenced to the soliton power is employed to capture and stabilize the solitons[53]. Shown in Fig. 2d are the optical spectra of solitons pumped at 1064 nm. These solitons are generated using the mode family whose dispersion is characterized in Fig. 2b. Due to the relatively low dispersion (small $D_2$), these solitons have a short temporal pulse width. Using the hyperbolic-secant-squared fitting method[2] (see orange and green curves in Fig. 2d), a soliton pulse width of 52 fs is estimated for the red spectrum. By increasing the soliton power (blue spectrum) the soliton can be further compressed to 44 fs, which corresponds to a duty cycle of 0.09% at the 20 GHz repetition rate. Finally, the inset in Fig. 2d shows the electrical spectrum of the photo-detected soliton pulse stream. Besides confirming the repetition frequency, the spectrum is very stable with excellent signal-to-noise ratio (SNR) greater than 70 dB at 1 kHz resolution bandwidth.

**Soliton generation at 778 nm**. As the operational wavelength shifts further toward the visible band, normal material dispersion increases. To generate solitons at 778 nm an additional dispersion engineering method, TM1-TE2 mode hybridization, is therefore added to supplement the geometrical dispersion control. The green band region in Fig. 1b gives the oxide thicknesses and wavelengths where this hybridization is prominent. Polarization mode hybridization is a form of mode coupling-induced

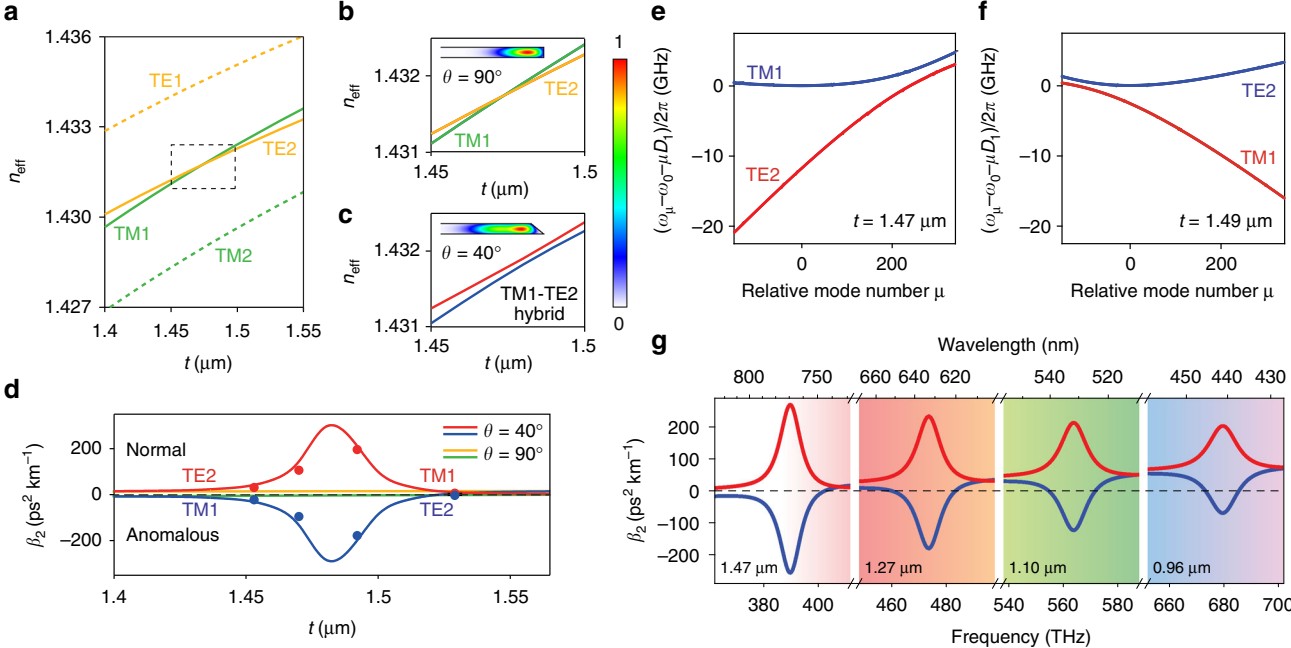

**Fig. 3** Dispersion engineering and soliton generation at 778 nm. **a** Calculated effective indices $n_{eff}$ for TE1, TE2, TM1, and TM2 modes at 778 nm plotted vs. thickness for a silica resonator with reflection symmetry (i.e., $\theta = 90°$). The TM1 and TE2 modes cross each other without hybridization. **b** Zoom-in of the dashed box in **a**. **c** As in **b** but for a resonator with $\theta = 40°$. An avoided crossing of TM1 and TE2 occurs due to mode hybridization. Insets of **b**, **c** show simulated mode profiles (normalized electric field) in resonators with $\theta = 90°$ and $\theta = 40°$, respectively. The color bar is shown to the right. **d** Calculated GVD of the two modes. For the $\theta = 40°$ case, hybridization causes a transition in the dispersion around the thickness 1.48 μm. The points are the measured dispersion values. **e**, **f** Measured relative mode frequencies of the TM1 and TE2 mode families vs. relative mode number μ for devices with $t = 1.47$ μm and $t = 1.49$ μm. **g** Calculated total second-order dispersion vs. frequency (below) and wavelength (above) at four different oxide thicknesses (number in lower left of each panel). Red and blue curves correspond to the two hybridized mode families. Anomalous dispersion is negative and shifts progressively to bluer wavelengths as thickness decreases. Background color gives the approximate corresponding color spectrum

dispersion control[22,38,39,54]. The coupling of the TM1 and TE2 modes creates two hybrid mode families, one of which features strong anomalous dispersion. This hybridization is caused when a degeneracy in the TM1 and TE2 effective indices is lifted by a broken reflection symmetry of the resonator[55]. The wavelength at which the degeneracy occurs is controlled by the oxide thickness and determines the soliton operation wavelength. Finite element method simulation in Fig. 3a shows that at 778 nm the TM1 and TE2 modes are expected to have the same effective index at the oxide thickness 1.48 μm when the resonator features a reflection symmetry through a plane that is both parallel to the resonator surface and that lies at the center of the resonator. Such a symmetry exists when the resonator has vertical sidewalls or equivalently a wedge angle $\theta = 90°$ (note: the wet-etch process used to fabricate the wedge resonators does not support a vertical side wall). A zoom-in of the effective index crossing is provided in Fig. 3b. In this reflection symmetric case, the two modes cross in the effective-index plot without hybridization. However, in the case of $\theta = 40°$ (Fig. 3c), the symmetry is broken and the effective index degeneracy is lifted. The resulting avoided crossing causes a sudden transition in the GVD as shown in Fig. 3d, and one of the hybrid modes experiences enhanced anomalous dispersion.

To verify this effect, resonators having four different thicknesses ($\theta = 40°$) were fabricated and their dispersion was characterized using the same method as for the 1064 nm soliton device. The measured second-order dispersion values are plotted as solid circles in Fig. 3d and agree with the calculated values given by the solid curves. Figure 3e, f shows the measured relative mode frequencies vs. mode number of the two modes for devices with $t = 1.47$ μm and $t = 1.49$ μm. As before, upward curvature in the data indicates anomalous dispersion. The dominant polarization component of the hybrid mode is also indicated on both

mode-family branches. The polarization mode hybridization produces a strong anomalous dispersion component that can compensate normal material dispersion over the entire band. Moreover, the tuning of this component occurs over a range of larger oxide thicknesses for which it would be impossible to compensate material dispersion using geometrical control alone. To project the application of this hybridization method to yet shorter soliton wavelengths, Fig. 3g summarizes calculations of second-order dispersion at a series of oxide thicknesses. At a thickness close to 1 micron, it should be possible to generate solitons at the blue end of the visible spectrum. Moreover, wedge resonators having these oxide film thicknesses have been fabricated during the course of this work. They are mechanically stable with respect to stress-induced buckling[56] at silicon undercut values that are sufficient for high-Q operation.

For soliton generation, the microresonator is pumped at 778 nm by frequency-doubling a CW ECDL operating at 1557 nm (see Fig. 4a). The 1557 nm laser is modulated by a quadrature phase-shift keying (QPSK) modulator for frequency-kicking[57] and then amplified by an erbium-doped fiber amplifier (EDFA). The amplified light is sent into a periodically poled lithium niobate (PPLN) device for second-harmonic generation. The frequency-doubled output pump power at 778 nm is coupled to the microresonator using a tapered fiber. The pump power is typically about 135 mW. The soliton capture and locking method was again used to stabilize the solitons[53]. A zoom-in of the TM1 mode spectrum for $t = 1.47$ μm with a fit that includes third-order dispersion (red curve) is shown in Fig. 4b. The impact of higher-order dispersion on dissipative soliton formation has been studied[58,59]. In the present case, the dispersion curve is well suited for soliton formation. The optical spectrum of a 778 nm pumped soliton formed on this mode family is shown in Fig. 4c. It features a

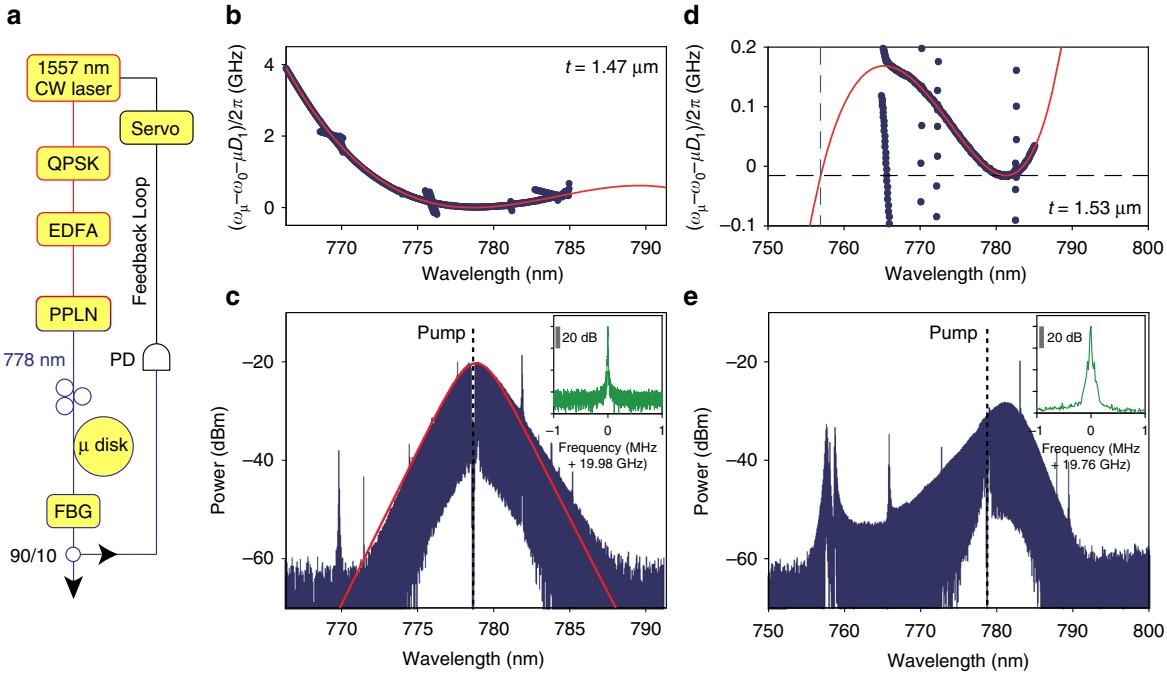

**Fig. 4** Solition generation at 778 nm. **a** Experimental setup for soliton generation. A 1557 nm tunable laser is sent to a QPSK modulator to utilize frequency-kicking[57] and is then amplified by an EDFA. Then, a PPLN waveguide frequency doubles the 1557 nm input into 778 nm output. The 778 nm pump light is coupled to the resonator for soliton generation. A servo loop is used to maintain pump locking[53]. **b** Measured relative mode frequencies of the TM1 mode family vs. wavelength for devices with $t = 1.47$ μm. A number of crossing mode families are visible. The red curve is a numerical fit using $D_2/2\pi = 49.8$ kHz and $D_3/2\pi = 340$ Hz. **c** Optical spectrum of a 778 nm soliton generated using the device measured in **b** with pump line indicated by the dashed vertical line. The red curve is a spectral fitting which reveals a pulse width of 145 fs. Most of the spurs in the spectrum correspond to the mode crossings visible in **b**. Inset shows the electrical spectrum of the detected soliton pulse stream. The resolution bandwidth is 1 kHz. **d** Measured relative mode frequencies of the TE2 mode family vs. wavelength for devices with $t = 1.53$ μm. The red curve is a fit with $D_2/2\pi = 4.70$ kHz and $D_3/2\pi = -51.6$ Hz. **e** Optical spectrum of a soliton generated using the device measured in **d** with pump line indicated as the dashed vertical line. A dispersive wave is visible near 758 nm. Inset shows the electrical spectrum of the detected soliton pulse stream. The resolution bandwidth is 1 kHz

temporal pulse width of 145 fs as derived from a sech$^2$ fit (red curve). The electrical spectrum of the photo-detected soliton stream is provided in the inset in Fig. 4c and exhibits high stability.

Figure 4d gives the measured mode spectrum and fitting under conditions of slightly thicker oxide ($t = 1.53$ μm). In this case, the polarization of the hybrid mode more strongly resembles the TE2 mode family. The overall magnitude of second-order dispersion is also much lower than for the more strongly hybridized soliton in Fig. 4b, c. The corresponding measured soliton spectrum is shown in Fig. 4e and features a dispersive wave near 758 nm. The location of the wave is predicted from the fitting in Fig. 4d (see dashed vertical and horizontal lines). The dispersive wave exists in a spectral region of overall normal dispersion, thereby illustrating that dispersion engineering can provide a way to further extend the soliton spectrum toward the visible band. As an aside, the plot in Fig. 4d has incorporated a correction to the FSR ($D_1$) so that the soliton line is given as the horizontal dashed black line. This correction results from the soliton red spectral shift relative to the pump that is apparent in Fig. 4e. This shift is a combination of the Raman self shift[49,50] and some additional dispersive wave recoil[3]. Finally, the detected beat note of the soliton and dispersive wave is shown as the inset in Fig. 4e. It is overall somewhat broader than the beatnote of the other solitons, but is nonetheless quite stable.

## Discussion
We have demonstrated soliton microcombs at 778 and 1064 nm using on-chip high-$Q$ silica resonators. Material-limited normal

dispersion, which is dominant at these wavelengths, was compensated by using geometrical dispersion through control of the resonator thickness and wedge angle. At the shortest wavelength, 778 nm, mode hybridization was also utilized to achieve anomalous dispersion while maintaining high optical $Q$. These results are the shortest wavelength soliton microcombs demonstrated to date. Moreover, the hybridization method can be readily extended so as to produce solitons over the entire visible band. The generated solitons have pulse repetition rates of 20 GHz at both wavelengths. Such detectable and electronics-compatible repetition rate soliton microcombs at short wavelengths have direct applications in the development of miniature optical clocks[9,22,23] and potentially optical coherence tomography[24–26]. Also, any application requiring low-power near-visible mode-locked laser sources will benefit. The same dispersion control methods used here should be transferable to silica ridge resonator designs that contain silicon nitride waveguides for on-chip coupling to other photonic devices[47]. Dispersive-wave generation at 758 nm was also demonstrated. It could be possible to design devices that use solitons formed at either 778 or 1064 nm for dispersive-wave generation into the visible and potentially into the ultra-violet as has been recently demonstrated using straight silica waveguides[42].

**Data availability**. The data that support the plots within this paper and other findings of this study are available from the corresponding author upon reasonable request.

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

## Acknowledgements

The authors thank Scott Diddams and Andrey Matsko for helpful comments on this work. The authors gratefully acknowledge the Defense Advanced Research Projects Agency under the ACES program (Award No. HR0011-16-C-0118) and the SCOUT

program (Award No. W911NF-16-1-0548). The authors also thank the Kavli Nanoscience Institute.

## Author contributions

S.H.L., D.Y.O., Q.-F.Y., B.S., H.W. and K.V. conceived the experiment. S.H.L. fabricated devices with assistance from D.Y.O., B.S., H.W. and K.Y.Y. D.Y.O., Q.-F.Y., B.S. and H.W. tested the resonator structures with assistance from S.H.L., K.Y.Y., Y.H.L. and X.Y. S.H.L., D.Y.O., Q.-F.Y., B.S., H.W. and X.L. modeled the device designs. All authors analyzed the data and contributed to writing the manuscript.

## Additional information

**Competing interests:** The authors declare no competing financial interests.

