## [Peer Review File · Nature Communications]

Reviewers' comments:

Reviewer #1 (Remarks to the Author):

This paper demonstrated a visible frequency comb based on the silica micro-disk resonator with high Q factor, which decreased the threshold of the pump power. The physical process is four-wave-mixing. The dispersion of the TM mode in visible range is controlled to anomalous by changing the shape the edge of the micro-cavity. It is helpful for optical clock cause the rubidium atom lines are visible. In addition, the stable comb can be realized if locking the comb line to a gas cell. So, this is interesting work, and there are some questions:

1. For visible wavelength, the micro-cavity is thinner (1.5 micron at 778nm). Is it very difficult to control the dispersion by changing the shape of the edge? Is the dispersion sensitive to wedge angle? In this experiment, the tapered fiber is used to couple pump into the micro-cavity. The thinness of the micro-cavity is 1.5 micron, and does it affect the stability of the comb cause the coupling condition is difficultly controllable. If so, I think this kind of micro-cavity is not best choice for visible comb.

2. What is the stable range of pump detuning for stable comb output? How long you can keep the stable mode-locking comb by using servo?

3. The low amplitude noise means the mode-locking state, but the autocorrelation can directly give pulse information. Could you measure the autocorrelation of the visible comb?

4. As shown in Fig.1, the comb spectrum of 1064nm is different with 1550nm and 778nm. What is the reason? Strong ASE from YDFA?

5. We can see that the dispersion $D^2/2\pi=3.3\text{KHz}$ is very small in Fig. 2b. What is the resolution of the Mach-Zehnder interferometer? The results are average?

6. For the optical clock, the stability is key point. I think that it will be better if the authors can lock the visible comb to gas cell for more stable comb. And I suggest that the authors can realize the visible comb using more integrated structure like Ref. 45.

In my opinion, the manuscript can be published after the authors have considered these comments.

Reviewer #2 (Remarks to the Author):

This paper written by S. H. Lee et al reports on soliton microcomb generation at IR and NIR wavelength, achieved by controlling the dispersion of the WGM microcavity. I think this paper presents an important step toward the generation of the soliton comb in a microcavity at shorter wavelength. Although the idea on controlling the geometric dispersion is not new, the tuning of the dispersion by the use of TM-TE coupling is nice. Also, the obtained result (achieving the soliton comb at 1064 and 780 nm) is having impact, so I support this paper to appear in Nature Communications. However, the effect of the TM-TE coupling is not clearly shown as claimed by the authors. So this point must be clarified.

1. According to the plot III in Fig. 1b, the system exhibits already small analogous dispersion even without the presence of TE-TM coupling. Therefore the effect of TE-TM coupling to the comb generation is not clear. I would love to know whether the authors obtained comb for the cavity shown in Fig. 3b, where not TE-TM coupling occurs.

2. I would like to see the β_2 curve for the structure with $\theta=90$ degree in Fig. 3d. Then we can directly compare the dispersions for $\theta=90$ and 40. I think this is essential to claim that the TE-TM effect is sufficiently affecting the dispersion.

3. Related to the previous point, I would like to see the curves for different wavelength. If I understand correctly, the curves in Fig. 3d are for the wavelength close at the pump. How about at the edge of the spectrum (for instance the curves for relative mode number 200)? I assume the peak of the curve shifts towards a smaller t for a shorter wavelength mode. I think this information is important, in order to know the information on the possible bandwidth of the comb generation.

In addition, the following points must be clarified.

4. The authors should add more plots in Fig. 3d. Three plots are not sufficient. The data appears me that the dispersion is monotonously increasing (or decreasing) rather than exhibiting a peak. I would like to see data points in between 1.47 and 1.49 μm . If this is difficult, why not 1.5 μm ?
5. What is the reason that you do not observe Raman shift (wavelength shift of the comb envelope towards lower frequency) in III in Fig. 1e
6. I think 778 nm is rather NIR than visible. I suggest the authors to change the title.
7. I see strong mode coupling close at the pump in Fig. 2b. Isn't it difficult to obtain smooth comb with such strong peak close at the pump? What is the reason that you do not suffer from this inter-mode coupling in Fig. 2b.

Reviewer #3 (Remarks to the Author):

The paper by Lee and colleagues reports on the generation of bright temporal dissipative cavity solitons in microresonators at wavelength of 1550 nm, 1064 nm and 778 nm. The latter represents the shortest temporal dissipative soliton generated so far in a microresonator and the first one in the visible wavelength regime (VIS \sim 400-800 nm).

The challenge in accomplishing soliton formation at 1064 nm and 780 nm was overcoming the 'dispersion barrier' imposed by increasingly normal material dispersion towards shorter wavelength (as bright solitons require anomalous dispersion).

In order to achieve anomalous dispersion the authors employ both geometrical dispersion engineering (for demonstration at 1064 nm) as well as additional mode-hybridization effects between TE and TM modes (for demonstration at 780 nm). Importantly, all three cases (1550 nm, 1064 nm and 778 nm) are based on the same technology platform, which illustrates the successful dispersion engineering (to the extent required for soliton generation) over a factor of two in optical wavelength. Another important aspect is that it was possible to maintain the same and electronically accessible soliton repetition rate of 20 GHz in all three cases.

Extending the operating wavelength range of microresonator solitons beyond initial demonstrations in the near-infrared towards visible and mid-infrared is potentially enabling a variety of novel applications e.g. in the bio-medical domain. As such the present demonstration of microresonator soliton generation at the edge of the visible regime is a novel and relevant to the field of microresonators and beyond. The presented data are of high quality and conclusively support the manuscript that is written in a clear manner.

Before I can recommend the manuscript without reservation for publication I would like to bring up the following points (in random order) for consideration by the authors. These points do not question the general novelty and importance of the work but are intended to help improving the manuscript:

- While not immediately related to bright solitons, the authors could include a comment on dark soliton or dark pulse generation, which provides an alternative method of creating microresonator frequency comb in the normal dispersion regime (Xue et al., Nature Photonics 9, 594, 2015).
- While threshold power levels are mentioned it is not clear whether they refer to the parametric threshold power or the threshold power for soliton generation (both notions seem to appear in the manuscript). Also it is not clear whether these power values specify the power in the tapered fiber or the coupled power when the threshold is reached.
- Besides threshold power levels, it would be interesting to know how much power in the tapered fiber was required to achieve the shown soliton spectra and what the coupling efficiency was in the respective modes (especially in the 778 nm case). It is clear that this first demonstration has not necessarily been optimized for efficiency or ideal coupling to the resonator; nevertheless the power

levels (in the tapered fiber) as well as the coupling efficiency are important characteristics of the setup and should be reported along with the generated optical spectra.

- Dispersion engineering via waveguide diameter, very similar to the 'thickness' parameter here, has already been employed in early work on microresonators (Del'Haye et al., PRL 107, 063901, 2011). While this work was not aiming at soliton formation the authors might want to consider citing this work to better put their work into context.

- The discussion in the manuscript focuses on whether the dispersion is anomalous or normal (sign of β_2 or D_2). It would be good if the authors could also discuss the (absolute) value of D_2 and the contribution of D_3 (as visible in Fig. 3g), which both supposedly increase width reduced operating wavelength. Estimating the values of D_2 and D_3 as well as the expected soliton bandwidth would be of interest, in particular as the authors suggest that the results could even extend across the visible into the ultraviolet bands. Helpful references in this context could be a study on the tolerance of solitons against non-zero D_3 (Herr et al., PRL 113, 123901, 2014) and universal scaling laws for e.g. the 3 dB bandwidth (Coen et al., Optics Letters 38, 11, 2013).

- Again related to further extending the results into the visible and potentially the UV bands: Can the authors comment on how thin the disks could be fabricated and operated before running into mechanical problems (vibrations or even collapse of the structure)?

- Using the mode hybridization is a novel and interesting approach. Can the authors comment on the achievable spectral bandwidth of this approach (which must be limited as the approximate mode degeneracy is probably restricted to a certain wavelength interval)?

- The dispersion measurements in Fig. 2a, 3d show error bars. I assume this is a result from the parabolic fit of the resonance frequencies. Can the authors comment on how the error bars were obtained in the presence of the outliers due to mode crossings?

- The RF beatnotes are narrow and show an impressive signal-to-noise ratio. It would however be interesting to choose a smaller scale e.g. 1 MHz span instead of 8 MHz, so that the small sidebands are better visible. Can the authors speculate on the origin of these noise (?) sidebands?

- As mentioned the manuscript is written in a clear manner but several minor language issues (plural/singular, misplaced words etc.) should be corrected.

Reviewer #1 (Remarks to the Author):

This paper demonstrated a visible frequency comb based on the silica micro-disk resonator with high Q factor, which decreased the threshold of the pump power. The physical process is four-wave-mixing. The dispersion of the TM mode in visible range is controlled to anomalous by changing the shape the edge of the micro-cavity. It is helpful for optical clock cause the rubidium atom lines are visible. In addition, the stable comb can be realized if locking the comb line to a gas cell. So, this is interesting work, and there are some questions:

1. For visible wavelength, the micro-cavity is thinner (1.5 micron at 778nm). Is it very difficult to control the dispersion by changing the shape of the edge? Is the dispersion sensitive to wedge angle? In this experiment, the tapered fiber is used to couple pump into the micro-cavity. The thinness of the micro-cavity is 1.5 micron, and does it affect the stability of the comb cause the coupling condition is difficultly controllable. If so, I think this kind of micro-cavity is not best choice for visible comb.

Reply: Thanks for the comment. In fact, we have an ability to precisely control and predict the dispersion of fabricated devices. The dispersion of the wedge resonators are largely determined by the two factors : the thickness and the wedge angle. Precise microfabrication provides excellent control of these parameters (See Ref 41 and 46). To illustrate this control we have provided measurements and simulation data of dispersion versus thickness in figure 2a along with banded regions that should be effect of wedge angle variation from 30 - 40 degrees. Our process control is typically of order several degrees. Thickness control is extremely good because the oxidation occurs over 43 hours. To clarify we have added a note in the text indicating that the oxidation is calibrated. The coupling condition is also very controlled. For the 778 nm experiment, we use a commercial single mode fiber 780HP to fabricate the taper with a minimum width around 1um. The length of the the thinnest part of the taper is around 2 mm and the transmission is usually above 90%. Since the width of the taper continuously changes around the taper center, we tune the position on the taper from which the light is coupled into the microcavity to approach the phase-matching condition. Using this method, we can achieve critical coupling condition for the resonators of various thicknesses with only one tapered fiber. Besides a stable coupling condition, the feedback loop which locks the pump laser frequency to a certain soliton power further allows long-term operation of the soliton (see Ref 5). In this experiment, both 1 um soliton and 778 nm soliton remained locked several hours until we turn off the pump laser. We are also working on a fully integrated visible soliton system on-chip which can eventually eliminate the need for tapered fiber coupling.

2. What is the stable range of pump detuning for stable comb output? How long you can keep the stable mode-locking comb by using servo?

Reply: The pump detuning for stable soliton operation depends on the pump power, but it is in the range of 10 cavity linewidths (tens of MHz in our case). The soliton comb can be stably operated for hours using the capture lock method (ref 53) until the pump laser was turned off. We have previously reported a record of soliton parameters over the duration of the mode-locking (see X. Yi et al. *Optica* 2, 1078 (2015) for a 24-hour measurement of a 1550 nm soliton - this is ref. 5 in this paper).

3. The low amplitude noise means the mode-locking state, but the autocorrelation can directly give pulse information. Could you measure the autocorrelation of the visible comb?

Reply: We are not set up to measure autocorrelation at these shorter wavelengths. However, we have measured autocorrelation for 1550 nm soliton combs generated in silica wedge resonators similar to those used in this paper (X. Yi et al. *Optica* 2, 1078 (2015) - ref. 5 in this paper). Significantly those measurements have confirmed the close agreement between the measured autocorrelation pulse width and the pulsewidth computed using the hyperbolic secant shape of the soliton spectrum. We are thus very confident in the prediction of pulsewidth provided in the paper by using the hyperbolic secant envelope provided in the data. We have included text in the manuscript that makes clear how we are calculating the pulsewidth and also reference the appropriate paper in case readers would like to check.

4. As shown in Fig. 1, the comb spectrum of 1064nm is different with 1550nm and 778nm. What is the reason? Strong ASE from YDFA?

Reply: Thanks for the comment. The difference is caused by the resolution of the OSA in frequency units, which decreases with decreasing wavelength. Therefore, the spectrum of 1064 nm has a lower contrast compared with the spectrum of 1550 nm. On the other hand, the 778 nm comb was measured as second-order diffraction at 1550 nm in the OSA and accordingly has a better resolution. We have added a comment in the revision to clarify this point.

5. We can see that the dispersion $D^2/2\pi=3.3\text{KHz}$ is very small in Fig. 2b. What is the resolution of the Mach-Zehnder interferometer? The results are average?

Reply: The dispersion is obtained by parabolically fitting the mode family spectrum over a large wavelength span to reduce the measurement error. The FSR of the MZI is around 40 MHz, which is calibrated to $<10^{-5}$ accuracy. We have added a sentence to indicate how the measurement was performed and to give the characteristics of the Mach-Zehnder interferometer.

6. For the optical clock, the stability is key point. I think that it will be better if the authors can lock the visible comb to gas cell for more stable comb. And I suggest that the authors can realize the visible comb using more integrated structure like Ref. 45.

Reply: Thanks for the suggestion. We have included a comment along these lines (i.e. ref 45 - now ref. 47) in the concluding paragraph of our manuscript. Furthermore, in the future we are working with other groups to ultimately lock this device to a Rb gas cell as suggested by the referee.

In my opinion, the manuscript can be published after the authors have considered these comments.

Reply: We thank the reviewer for their comments which have improved our manuscript.

Reviewer #2 (Remarks to the Author):

This paper written by S. H. Lee et al reports on soliton microcomb generation at IR and NIR wavelength, achieved by controlling the dispersion of the WGM microcavity. I think this paper presents an important step toward the generation of the soliton comb in a microcavity at shorter wavelength. Although the idea on controlling the geometric dispersion is not new, the tuning of the dispersion by the use of TM-TE coupling is nice. Also, the obtained result (achieving the soliton comb at 1064 and 780 nm) is having impact, so I support this paper to appear in Nature Communications. However, the effect of the TM-TE coupling is not clearly shown as claimed by the authors. So this point must be clarified.

1. According to the plot III in Fig. 1b, the system exhibits already small analogous dispersion even without the presence of TE-TM coupling. Therefore the effect of TE-TM coupling to the comb generation is not clear. I would love to know whether the authors obtained comb for the cavity shown in Fig. 3b, where not TE-TM coupling occurs.

Reply: Thanks for the comment. Indeed some modes can exhibit small amount of anomalous dispersion in the absence of the TE-TM coupling. However, as shown in the revised Fig. 3d, the dispersion is very close to zero and therefore making the soliton generation difficult, because the system is more sensitive to distortions in mode family dispersion.

Also, we do not have means to fabricate the devices with $\theta = 90$ degree for dispersion measurement because of the nature of the wet-chemical etching process used to create the high-Q silica wedge devices. As a result, the TE - TM mode coupling is always present to some degree in our samples. We have added a sentence to indicate that the 90 degree sidewall case is not possible with the current etch process.

2. I would like to see the β_2 curve for the structure with $\theta=90$ degree in Fig. 3d. Then we can directly compare the dispersions for $\theta=90$ and 40. I think this is essential to claim that the TE-TM effect is sufficiently affecting the dispersion.

Reply: Thanks for the comment. We added the calculated dispersion of $\theta=90$ degree resonators in Fig. 3d (horizontal line) which makes clear that the TE-TM coupling greatly increases the amount of dispersion at a certain thickness. . As noted above (and now in a comment in the manuscript) we do not have means to fabricate high-Q devices with $\theta = 90$ degree for dispersion measurement because of the nature of the wet-chemical etching process used to create the high-Q silica wedge devices. We agree that this would be an interesting measurement. On the other hand, the agreement of measured dispersion versus thickness with modeling provides strong evidence that the mode hybridization is providing the intended dispersion. Also, we note that additional resonator thicknesses have been added (per your comment below) to the data which further confirm the effect.

3. Related to the previous point, I would like to see the curves for different wavelength. If I understand correctly, the curves in Fig. 3d are for the wavelength close at the pump. How about at the edge of the spectrum (for instance the curves for relative mode number 200)? I assume the peak of the curve shifts towards a smaller t for a shorter wavelength mode. I think this information is important, in order to know the information on the possible bandwidth of the comb generation.

Reply: Thanks for the comment. We have added a new simulation in figure 3g that directly addresses the referee's comment. It plots the second order dispersion versus wavelength at a series of thicknesses. The bandwidth of the hybridization effect can be directly seen in these plots.

In addition, the following points must be clarified.

4. The authors should add more plots in Fig. 3d. Three plots are not sufficient. The data appears me that the dispersion is monotonously increasing (or decreasing) rather than exhibiting a peak. I would like to see data points in between 1.47 and 1.49 μm . If this is difficult, why not 1.5 μm ?

Reply: Thanks for the suggestion. We added more data points. They agree well with the simulation and importantly verify that the effect is not monotonic.

5. What is the reason that you do not observe Raman shift (wavelength shift of the comb envelope towards lower frequency) in III in Fig. 1e

Reply: The Raman shift for the 778 nm soliton microcomb in Fig. 1e and Fig. 4c is minimal because of relatively large pulse width (145 fs derived from sech^2 -fit) and small spectral bandwidth. The calculated Raman SSFS for this case using the formula from Ref. 50 is 0.3 nm. We added a sentence to clarify this point.

6. I think 778 nm is rather NIR than visible. I suggest the authors to change the title.

Reply: Although the pump wavelength 778 nm is not within the visible range, a portion of the soliton frequency comb is actually visible. Indeed, we can observe red light emitting from the microcavity using bare eyes (also see fig. 1d). Furthermore, we have added a new result (Fig. 4e) showing an even broader soliton spectrum whose wavelength can reach as low as 755 nm.

7. I see strong mode coupling close at the pump in Fig. 2b. Isn't it difficult to obtain smooth comb with such strong peak close at the pump? What is the reason that you do not suffer from this inter-mode coupling in Fig. 2b.

Reply: Although the mode crossing is close to the pump, it only affects one mode with minor distortions (see the optical spectra in Fig. 2d). Moreover, the Raman effect shifts the soliton envelope center towards longer wavelength and thereby further minimizes the influence of this mode crossing to the soliton. Therefore, the soliton is stable and easy to generate.

We thank the reviewer for their comments which have improved our manuscript.

Reviewer #3 (Remarks to the Author):

The paper by Lee and colleagues reports on the generation of bright temporal dissipative cavity solitons in microresonators at wavelength of 1550 nm, 1064 nm and 778 nm. The latter represents the shortest temporal dissipative soliton generated so far in a microresonator and the first one in the visible wavelength regime (VIS ~ 400-800 nm). The challenge in accomplishing soliton formation at 1064 nm and 780 nm was overcoming the 'dispersion barrier' imposed by increasingly normal material dispersion towards shorter wavelength (as bright solitons require anomalous dispersion). In order to achieve anomalous dispersion the authors employ both geometrical dispersion engineering (for demonstration at 1064 nm) as well as additional mode-hybridization effects between TE and TM modes (for demonstration at 780 nm). Importantly, all three cases (1550 nm, 1064 nm and 778 nm) are based on the same technology platform, which illustrates the successful dispersion engineering (to the extent required for soliton generation) over a factor of two in optical wavelength. Another important aspect is that it was possible to maintain the same and electronically accessible soliton repetition rate of 20 GHz in all three cases.

Extending the operating wavelength range of microresonator solitons beyond initial demonstrations in the near-infrared towards visible and mid-infrared is potentially enabling a variety of novel applications e.g. in the bio-medical domain. As such the present demonstration of microresonator soliton generation at the edge of the visible regime is a novel and relevant to the field of microresonators and beyond. The presented data are of high quality and conclusively support the manuscript that is written in a clear manner.

Before I can recommend the manuscript without reservation for publication I would like to bring up the following points (in random order) for consideration by the authors. These points do not question the general novelty and importance of the work but are intended to help improving the manuscript:

- While not immediately related to bright solitons, the authors could include a comment on dark soliton or dark pulse generation, which provides an alternative method of creating microresonator frequency comb in the normal dispersion regime (Xue et al., Nature Photonics 9, 594, 2015).

Reply: Thanks for the comment. We have included this citation and added a comment in the revision.

- While threshold power levels are mentioned it is not clear whether they refer to the parametric threshold power or the threshold power for soliton generation (both notions seem to appear in the manuscript). Also it is not clear whether these power values specify the power in the tapered fiber or the coupled power when the threshold is reached.

Reply: The threshold power mentioned in Fig. 1c refers to parametric oscillation threshold. These values are the power launched in the tapered fiber while the resonator is critically coupled. We have clarified it in the revision.

- Besides threshold power levels, it would be interesting to know how much power in the tapered fiber was required to achieve the shown soliton spectra and what the coupling efficiency was in the respective modes (especially in the 778 nm case). It is clear that this first demonstration has not necessarily been optimized for efficiency or ideal coupling to the resonator; nevertheless the power levels (in the tapered fiber) as well as the coupling efficiency are important characteristics of the setup and should be reported along with the generated optical spectra.

Reply: For 1 μ m soliton, the minimum pump power is 100mW, while for 778 nm soliton, it is 135 mW. We have added the numbers in the revision.

- Dispersion engineering via waveguide diameter, very similar to the 'thickness' parameter here, has already been employed in early work on microresonators (Del'Haye et al., PRL 107, 063901, 2011). While this work was not aiming at soliton formation the authors might want to consider citing this work to better put their work into context.

Reply: Thanks for the comment. We have added the citation.

- The discussion in the manuscript focuses on whether the dispersion is anomalous or normal (sign of β_2 or D_2). It would be good if the authors could also discuss the (absolute) value of D_2 and the contribution of D_3 (as visible in

Fig. 3g), which both supposedly increase width reduced operating wavelength. Estimating the values of D2 and D3 as well as the expected soliton bandwidth would be of interest, in particular as the authors suggest that the results could even extend across the visible into the ultraviolet bands. Helpful references in this context could be a study on the tolerance of solitons against non-zero D3 (Herr et al., PRL 113, 123901, 2014) and universal scaling laws for e.g. the 3 dB bandwidth (Coen et al., Optics Letters 38, 11, 2013).

Reply: We agree with the referee and have added a new fig. 3g which provides a simulation of the GVD versus wavelength at a series of oxide thicknesses. These plots provide information on the useful bandwidth of the hybridization effect and also illustrate the required thickness in order to extend the effect across the visible band. We have added the fitted D2 and D3 values in Fig. 4b and Fig. 4d. We have also added a comment in the text relating to higher order dispersion and have also added the citations suggested by the referee.

- Again related to further extending the results into the visible and potentially the UV bands: Can the authors comment on how thin the disks could be fabricated and operated before running into mechanical problems (vibrations or even collapse of the structure)?

Reply: As noted above, figure 3g has been added to show that a thickness of about 1 micron is required for soliton operation in the blue end of the visible spectrum. At the same time we have fabricated a series of disks with oxide thickness down to this value. They are mechanically stable with respect to silicon undercut to levels that can provide high Q operation. We have accordingly added a comment to this effect and also cited our earlier paper on stress buckling (Chen, Tong, Hansuek Lee, and Kerry J. Vahala. "Thermal stress in silica-on-silicon disk resonators." *Applied Physics Letters* 102.3 (2013): 031113.).

- Using the mode hybridization is a novel an interesting approach. Can the authors comment on the achievable spectral bandwidth of this approach (which must be limited as the approximate mode degeneracy is probably restricted to a certain wavelength interval)?

Reply: As noted above, we have added fig. 3g which answers this question.

- The dispersion measurements in Fig. 2a, 3d show error bars. I assume this is a result from the parabolic fit of the resonance frequencies. Can the authors comment on how the error bars were obtained in the presence of the outliers due to mode crossings?

Reply: Error bars were obtained by measuring dispersion of various samples for each thickness. Fitting errors are much smaller than sample variations and are therefore ignored. We have noted how error bars were determined in Fig. 2a. In Fig. 3d, the error bars were so small (absolute dispersion is much larger compared with Fig. 2a) that we have decided to omit them in favor of larger data points.

- The RF beatnotes are narrow and show an impressive signal-to-noise ratio. It would however be interesting to choose a smaller scale e.g. 1 MHz span instead of 8 MHz, so that the small sidebands are better visible. Can the authors speculate on the origin of these noise (?) sidebands?

Reply: Thanks for the comment. We have rescaled our plot as suggested by the reviewer. The sidebands near 10 kHz originate from the feedback loop, i.e., the pump laser piezo tuning bandwidth. We have added a comment to this effect in the figure caption.

- As mentioned the manuscript is written in a clear manner but several minor language issues (plural/singular, misplaced words etc.) should be corrected.

Reply: Thanks for the comment. We have made improvements in the revised manuscript to correct typographical errors.

We also thank the reviewer for their comments which have improved our manuscript.

REVIEWERS' COMMENTS:

Reviewer #1 only submitted Remarks to the Editor

Reviewer #2 (Remarks to the Author):

I am satisfied with the revision made by the authors. I have nothing more to add.
Congratulations!

Reviewer #3 (Remarks to the Author):

The authors have fully addressed all my previous comments and questions; from my perspective the manuscript can be published without any further delay.